# Measuring Zero-Dose Children: Reflections on Age Cohort Flexibilities for Targeted Immunization Surveys at the Local Level

**DOI:** 10.3390/vaccines12020195

**Published:** 2024-02-14

**Authors:** Gustavo C. Corrêa, Md. Jasim Uddin, Tasnuva Wahed, Elizabeth Oliveras, Christopher Morgan, Moses R. Kamya, Patience Kabatangare, Faith Namugaya, Dorothy Leab, Didier Adjakidje, Patrick Nguku, Adam Attahiru, Jenny Sequeira, Nancy Vollmer, Heidi W. Reynolds

**Affiliations:** 1Gavi, The Vaccine Alliance, Chemin du Pommier 40, Le Grand Saconnex, 1218 Geneva, Switzerland; 2International Centre for Diarrhoeal Disease Research, Bangladesh, 68 Shaheed Tajuddin Ahmed Sarani, Mohakhali, Dhaka 1212, Bangladeshtasnuva.wahed@icddrb.org (T.W.); 3Jhpiego, The Johns Hopkins University Affiliate, 1615 Thames Street, Baltimore, MD 21231, USAchristopher.morgan@jhpiego.org (C.M.); 4Infectious Diseases Research Collaboration (IDRC), Kampala P.O. Box 7475, Uganda; mkamya@idrc-uganda.org (M.R.K.); faithsentongo@gmail.com (F.N.); 5Department of Medicine, Makerere University, Kampala P.O. Box 7072, Uganda; 6GaneshAID, 143 Doc Ngu, Lieu Giai, Ba Dinh, Hanoi 152960, Vietnam; 7African Field Epidemiology Network (AFENET), 50 Haile Selassie St, Asokoro, Abuja 900103, Nigeria; 8The Geneva Learning Foundation (TGLF), Av. Louis-Casaï 18, 1209 Geneva, Switzerland; 9JSI Research & Training Institute, Inc. (JSI), 2733 Crystal Dr 4th Floor, Arlington, VA 22202, USA; nancy_vollmer@jsi.com

**Keywords:** zero-dose, equity, immunization, targeted surveys, measurement

## Abstract

Zero-dose (ZD) children is a critical objective in global health, and it is at the heart of the Immunization Agenda 2030 (IA2030) strategy. Coverage for the first dose of diphtheria–tetanus–pertussis (DTP1)-containing vaccine is the global operational indicator used to estimate ZD children. When surveys are used, DTP1 coverage estimates usually rely on information reported from caregivers of children aged 12–23 months. It is important to have a global definition of ZD children, but learning and operational needs at a country level may require different ZD measurement approaches. This article summarizes a recent workshop discussion on ZD measurement for targeted surveys at local levels related to flexibilities in age cohorts of inclusion from the ZD learning Hub (ZDLH) initiative—a learning initiative involving 5 consortia of 14 different organizations across 4 countries—Bangladesh, Mali, Nigeria, and Uganda—and a global learning partner. Those considerations may include the need to generate insights on immunization timeliness and on catch-up activities, made particularly relevant in the post-pandemic context; the need to compare results across different age cohort years to better identify systematically missed communities and validate programmatic priorities, and also generate insights on changes under dynamic contexts such as the introduction of a new ZD intervention or for recovering from the impact of health system shocks. Some practical considerations such as the potential need for a larger sample size when including comparisons across multiple cohort years but a potential reduction in the need for household visits to find eligible children, an increase in recall bias when older age groups are included and a reduction in recall bias for the first year of life, and a potential reduction in sample size needs and time needed to detect impact when the first year of life is included. Finally, the inclusion of the first year of life cohort in the survey may be particularly relevant and improve the utility of evidence for decision-making and enable its use in rapid learning cycles, as insights will be generated for the population being currently targeted by the program. For some of those reasons, the ZDLH initiative decided to align on a recommendation to include the age cohort from 18 weeks to 23 months, with enough power to enable disaggregation of key results across the two different cohort years. We argue that flexibilities with the age cohort for inclusion in targeted surveys at the local level may be an important principle to be considered. More research is needed to better understand in which contexts improvements in timeliness of DTP1 in the first year of life will translate to improvements in ZD results in the age cohort of 12–23 months as defined by the global DTP1 indicator.

## 1. Introduction

The Immunization Agenda 2030 (IA2030) places missed communities at the heart of its current strategy [1]. Those missed communities under multiple deprivations are considered clusters of zero-dose (ZD) children, systematically not reached by immunization programs. IA2030 defines ZD children as those who did not receive their first dose of a diphtheria–tetanus–pertussis (DTP1)-containing vaccine, and it uses DTP1 coverage to estimate ZD numbers [2]. The rationale for this indicator is that DTP1 is universally used in routine immunization programs across different countries and is usually administered at the first point of contact of communities with the health system, being recommended for infants as early as 6 weeks of age [3]. Gavi, the Vaccine Alliance supports countries to reach those ZD objectives and uses the same ZD definitions and indicators for its current strategy [4,5].

To track annual DTP1 coverage progress, global organizations rely on the World Health Organization (WHO) and United Nations International Children’s Emergency Fund (UNICEF) Estimates of National Immunization Coverage (WUENIC). WUENIC relies on country data officially reported to WHO and UNICEF by Member States. Data for WUENIC is generally sourced from administrative systems and surveys conducted at the national level [6].

In many countries, it can be challenging to have reliable coverage estimates based on administrative data. When administrative systems are used, DTP1 coverage is generally calculated by dividing the number of children receiving the vaccine during their first year of life by the estimated number of children who survived their first year of life. However, there are multiple data quality issues that can impact both the numerator and the denominator in this formula. Errors in numerators can be caused by suboptimal data collection and system tools, poor documentation practices, intentional falsification, and lack of reporting from non-governmental providers and there is also much uncertainty on denominators projections [7]. Both errors in numerators and denominators may be aggravated in systematically missed communities with high numbers of ZD children.

Probability-based household surveys may provide an alternative data source for coverage estimation, and WUENIC also relies on national survey data to estimate DTP1 coverage in many countries. For better comparability with administrative data, the estimate is generally based on the report of an annual cohort of children. Usually, surveys based DTP1 coverage estimates rely on information reported from caregivers of children aged 12–23 months, which is also the standard definition used for DTP1 coverage estimation from widely used global survey methodologies, such as the Demographic and Health Survey (DHS) [8] and the Multiple Indicator Cluster Survey (MICS) [9] methods. The main reason why DTP1 coverage estimates start from 12 months of life is because immunization coverage surveys ask about all antigens in the vaccination schedule which should be given by the time a child turns 1 year old. Therefore, measurement of DTP1 coverage starts with children 12 months of age and older, up to 23 months. This way, all children included in the survey cohort would have had the opportunity to receive all age-appropriate vaccinations across all antigens in the first year of life. Following the same logic, a second cohort of children aged 24–35 months may also be surveyed for measuring coverage of antigens administered during the second year of life, but this second cohort is generally not used to measure DTP1 coverage.

Having a clear international definition for measuring ZD children both from administrative systems and from surveys enables the use of existing data in a standardized way to track progress at a portfolio level for global programs. It also enables a common understanding of global level drivers and helps inform activities designed to improve health information system adjustments with key indicators in mind. It simplifies messages used in global communication materials and enables alignment of advocacy efforts, providing a clear direction to the global community. Indeed, national level surveys using DTP1 coverage estimates based on the 12–23-month age cohort have been very important for learning about ZD children distribution, association with multiple deprivations, and drivers at the global level [10,11,12,13,14,15].

Beyond those global use cases, learning and operational needs at the country level may require different measurement approaches and, where relevant, feasible, and affordable, targeted surveys at the local level may be particularly well placed to identify and gather relevant information on critical equity-related issues. They may be especially useful when targeted to selected communities under multiple deprivations, such as those living in urban slums, in hard-to-reach areas, who are nomadic, refugees or who have been displaced, or belong to ethnic minorities or religious closed communities, among many others.

Because they are targeted, they can offer critical insights on multiple deprivations affecting specific communities and highlight key enablers and drivers to immunization programs with a robustness that cannot be easily achieved with surveys powered at the national or regional level. This type of targeted evidence can be very useful to inform approaches towards other missed communities facing similar contexts.

They may also be a good method to validate the selection of missed communities to be prioritized by the immunization program. A critical assumption of the ZD agenda is that countries should target systematically missed communities and bring them towards full immunization and other primary healthcare services, but it may not be simple to ascertain if communities are systematically missed. The missed communities identified by the program may have never been registered in the local area administrative system, but they may have a different health seeking behavior and they could, for example, be immunized through health services in another area or through other private providers not reporting to the local administrative system. In those cases, they would not be systematically missed, just not registered by the local area health system and targeted surveys can generate clear evidence on this topic where administrative data may fail.

They can also be critical to support monitoring and evaluate programmatic impact in local areas, supplementing routinely collected data while relevant activities to improve data collection and quality in missed communities are rolled out. This may generate critical early evidence to inform adjustments in programmatic interventions and policies with key communities under multiple vulnerabilities in sight [16]. 

The ZD Learning Hub (ZDLH)—a ZD learning initiative engaging 5 different consortia of partners across 4 countries (Bangladesh, Mali, Nigeria, and Uganda) and at the global level and involving 14 different organizations [17], had a comprehensive discussion on how to better measure ZD children when targeted surveys are used at local level to respond to specific learning needs in a recent workshop. Targeted surveys at the local level have been proposed for the four ZDLH countries. Different methodologies and approaches, adapted to local contexts are proposed and will help answer contextually relevant research questions, but with some commonalities. Firstly, country ZDLH propose to assess the magnitude of the ZD problem in some local communities that have already been prioritized through a national level exercise using secondary data analysis and stakeholder consultations. Well-designed surveys using random sampling frames in some key areas are suggested to validate the country prioritization of key communities. Secondly, targeted surveys at the local level are proposed to better understand ZD drivers affecting childhood immunization in those missed communities to better tailor programmatic activities. Although drivers of ZD children can often be extracted from national level surveys, information on drivers affecting specific communities with higher numbers of ZD children may be inadequate or unavailable. In addition, indicators on other types of drivers using novel and useful tools, such as the behavioral and social driver (BeSD) tools [18], are often not included in traditional surveys among other specific components that can provide insights on specific demand-related barriers affecting specific communities. Finally, measuring the impact of specific interventions designed to reach ZD children in those specific communities is also an objective across countries. This is most often proposed to be achieved with the implementation of at least two rounds of surveys at the same area with trends over time.

All those objectives could be achieved with the traditional approach of using DTP1 coverage estimates from surveys based on the 12 to 23 months age cohort, but this approach can also bring some important gaps. In this article, we synthesize the discussions from the ZDLH group related to age cohort of inclusion in targeted surveys at the local level with the general objective of generating insights to improving methodological approaches for those surveys. We make the case that flexibility with operational definitions of ZD children—particularly related to the age cohort of inclusion across the first years of life—is an important principle to respond to local learning agendas needs. The ZDLH initiative has decided to align on a general recommendation to expand the age cohort for its targeted survey from 18 weeks to 23 months to better respond to key project objectives and research questions, and we also synthesize the reasons for this decision.

## 2. Key Considerations for Flexibilities of Age Cohort of Inclusion in Targeted Local Surveys

### 2.1. The Case for the Inclusion of Other Age Cohorts in the First Years of Life

Different countries may have different key concerns related to communities missed by immunization programs. Some may be more focused on reaching children with immunization in a timely manner and others will be concentrating efforts to reach children that may have been missed during a previous period of crisis, such as the COVID-19 pandemic.

Immunization coverage surveys following international standards collect and report DTP1 data from an annual cohort of children aged 12–23 months. This standard age range enables identification of systematically missed communities such as those who were not reached at the end of their first year of life tend to never be reached. However, it usually does not incorporate and generate insights on other relevant principles for children across other age cohorts.

One such relevant principle related to the first year of life is the concept of immunization timeliness. It refers to receiving vaccinations at the earliest appropriate age to confer optimal immunological protection to the child. For DTP1, WHO recommends vaccines from as early as 6 weeks of age and for the third dose of DTP vaccine (DTP3) the recommendation is as early as 14 weeks [3]. The timeliness concept is critical in the first year of life for multiple reasons. After a child is born, transplacental immunity quickly decreases, putting the infant at risk of death and disability from vaccine preventable disease at a time when they are particularly vulnerable [19]. If vaccines are provided too early or too closely spaced, it may not generate an adequate immunological response and reduce duration of protection. When vaccines are delayed, it increases the number of individuals susceptible to specific diseases, reducing herd immunity and exposing the community to circulating vaccine preventable diseases, also putting individuals with medical contraindications and reduced immunity at risk [20].

Generally, administrative data systems do not record the age by which the child has been immunized, thus not allowing the generation of insights on timeliness of immunization. Surveys, in most countries are the only available option when programs need to assess the first year of life. International survey methodologies such as DHS [8] and MICS [9] typically record the age of vaccine administration and, although their standard indicators do not include the immunization timeliness concept, they can and have been used to retrospectively estimate immunization timeliness in the first year of life at the national level across different studies [21,22]. However, in specific communities with high numbers of ZD children, assessing timeliness based on household surveys may be challenging as home-based records (HBR) tend not to be available, making it difficult to retrieve specific dates for vaccine administration and making it inadequate to retrospectively assess it. When the first-year cohort is included in a targeted local survey, it may enable the generation of more reliable and timely insights on immunization timeliness. It may also enable to focus on specific communities with higher number of ZD children, to understand key drivers for untimely immunization, which could be, in some cases, easily actioned by the program.

In addition to the first year of life, countries may also have the need to gather insights about older children and that may be particularly true in the current global context of post-pandemic recovery. There has been clear documentation that the COVID-19 pandemic affected health systems in many countries, which had different recovery speeds [23]. According to the most recent WUENIC release, some countries have not yet fully recovered [24]. WHO, UNICEF, Gavi, and IA2030 recently launched “The Big Catch Up” initiative to intensify efforts to catch up missed children during the pandemic at the global level [25], but there is poor evidence on the local impact of the pandemic in specific communities, what the key local drivers were, and what could work to address the situation. A targeted local survey that includes older age cohorts—beyond 23 months—could also provide good information to support interpretation of the COVID-19 reminiscent impact on ZD, understand recovery trends, and better inform programs in their current catch-up efforts.

### 2.2. The Case for Comparison of Different Age Cohorts in the Same Targeted Local Survey

Including multiple age cohorts in a single targeted local survey offers the possibility of comparisons of results and key ZD drivers across different cohort years. This may be relevant because each different age cohort will represent a different year of exposure to programmatic activities. Because most vaccines are administered in the first year of life, there is generally a linear correlation of a child’s age and the timing of program reach. In other words, the first year of life tends to represent results of the current programmatic year, the second year of life tends to represent results from a year ago, and so forth. Comparing those different age cohorts enables a better understanding of programmatic trends of coverage and drivers in targeted communities to better identify when children are systematically missed and also provides insights on how to better reach them. This may be especially useful to understand communities in dynamic contexts, such as when communities are submitted to a health system shock or on the introduction of a new intervention.

Understanding the immunization results across multiple years in communities is a key need of ZD programming, but focusing on a single cohort year may also not be enough to identify a systematic failure to reach them. A poor immunization performance in one programmatic year could be an outlier—the result of a specific health system shock that was atypical and will not be sustained over time. There are many reasons a community could be affected by time-bound health system shocks. Those could be localized shocks such as a local stock out of vaccines, a key cold chain equipment breakage, an outbreak of infectious disease, a natural disaster, an atypical severe weather event blocking road access, or a local conflict deflagration, among many others. They could also be the result of a national shock such as events of political instability, or a global shock such as global shortages of vaccines or the recent COVID-19 pandemic. If surveys are designed with multiple cohort years, they may be a powerful tool to support the identification of systematically missed communities, or those not affected only by time-bound shocks.

Once a community has been confirmed as systematically missed, there may still be a need to compare results and how the ZD drivers may be shifting across different cohort years. Surveys can measure coverage and also include questions about critical drivers and qualitative components across age cohorts, and this can highlight how well communities are reached and how determinants have changed over time. This may be a critical objective of the implementation research.

Including comparisons of results for the first year of life with other age cohorts in targeted local surveys may be particularly helpful for new ZD interventions. As programs are designed to reach systematically missed communities and as they get better in this objective, we should expect significant shifts in the determinants of ZD children in those communities in a very short timeframe. What could be mostly due to a chronic lack of access can quickly become a demand issue only some months after the introduction of an intervention, and this may have important implications for program adaptation.

The same logic applies to older age cohorts when health systems are recovering from time-bound shocks or crises. Comparing the results and drivers from older age cohorts, at a period when a crisis was hitting hard, with more recent ones when things have been more stable may provide convincing proof of the crisis impact in immunization results for a given community and provide insights on its recovery speed, building evidence for the potential relevance of targeted catch-up efforts. It can also generate useful insights into the design of those catch-up efforts, as it can better demonstrate how prevalent shock-related barriers have been and may still be present, and which programs may need to adjust current activities to better address them. In addition, it may generate evidence for other new interventions to increase the health system resiliency and ensure these crisis-related barriers will not have the same weight in future crisis events.

### 2.3. Practical Considerations

There are also a number of practical considerations that may need to be carefully dealt with when flexibilities with the age cohort of inclusion are being contemplated in targeted local surveys.

Firstly, when comparison of results and ZD drivers across cohort years using a single survey or when precise estimates for specific cohort years are critical objectives, there will likely be implications for sample size calculations and fieldwork planning.

The study will need to ensure that results for each cohort year are sufficiently reliable to enable meaningful statistical comparisons and conclusions. In those cases, a targeted local survey may need to have enough power to enable disaggregation and comparisons of the data by each relevant age cohort included in the original research questions. To achieve this objective, sample size calculations may need to be separately performed for each cohort year included but allowing simultaneous data collection using the same logistical structure. The lowest number of individuals to be included in the study sample must enable answering the research question that requires a larger sample size, focusing on few critical primary objectives.

WHO has developed useful guidance focused on clustered surveys, indicating methods for sample size calculation for different research questions and objectives that researchers can already build upon [26]. There is also sampling guidance for Lot Quality Assurance Sampling (LQAS) surveys, mainly for the integration of data from different programs in a single survey [27]. The calculation of sample sizes for local targeted surveys focused on ZD children, which tend to be non-clustered and, to use a single indicator across different cohort years, will likely make the task simpler regardless of the survey method. 

Despite of the need for a higher sample size in those cases, expanding to a larger age cohort and responding to different research questions or comparing age cohorts using a single survey may also simplify fieldwork, as it may require a substantially lower number of household visits to find eligible children; that is, those matching the wider age range for inclusion. The higher probability of finding children in the eligible age range in any single household visit would reduce the number of household visits required to fulfil the sample size needs.

Secondly, adding other age cohort years in the survey will have implications on recall bias which may affect data reliability and validity.

Generally, when immunization surveys are conducted, the vaccination history is preferably captured from documented evidence sources, such as HBR—and less often from health facility-based records (FBR). Very frequently, it will be based on survey respondent memory recall, especially when HBR are not available [28].

HBR are generally considered a more reliable source of vaccination history data, despite running some risks of containing errors such as incomplete recording, mis-recording or mismatch between children being surveyed and the card presented by the caregiver. However, in many countries with high numbers of ZD children, HBR are often not available [29] and FBR may be of very poor quality. This situation may be aggravated for surveys targeting missed communities with highest numbers of ZD children. In those settings, it is likely that vaccination history will often rely in recall.

Recall data are sometimes not correlated with HBR data and, in general, may have poor agreement with other sources [30]. Memory bias is frequent, as caregivers tend to over-report coverage due to social desirability bias, or they simply may be unable to remember the vaccination history with details. This concern may be intensified due to the growing complexity of the vaccination schedule.

When multiple age cohort years are included in a targeted survey focused on missed communities, researchers should pay special attention to the risk of memory bias from recall, which may affect different age cohorts in dissimilar ways. DTP1 coverage and the ZD concept may itself be less subject to memory bias than other later doses, as the caregiver will have a lower risk of not remembering a first dose of vaccine for a child than to be precise about the number of doses received across the schedule. However, it is reasonable to assume memory bias may play a larger role for older age groups as there will be a larger amount of time elapsed from the time the vaccine has been received and it will be less important for the first year of life as less time may have elapsed from the vaccine dose to the survey inquiry.

In addition, when repeated surveys are considered for estimating the impact of ZD interventions, it is also likely that DTP1 coverage may be overestimated in the first round due to recall bias and underestimated in the second round, assuming ZD interventions are rolled out and those missed communities are finally reached and HBR become more available as part of the intervention. This pattern of coverage overestimation being observed from recall and under estimation from HBR has been recently suggested by a recent and comprehensive systematic review [31]. This may reduce the measured treatment effect size and have other implications for survey design, sample size, and analytical plan. Those limitations need to be highlighted and data from those surveys must be interpreted with care.

Thirdly, when an impact assessment is a critical study objective, the inclusion of the first year of life in the age range of a targeted local survey may considerably simplify the study operationalization. In those cases, it may significantly decrease the sample size needed because the estimated treatment effect size will likely be higher. This is especially true in settings where the prevalence of ZD children is relatively low. Importantly, it will also decrease the time needed to detect impact. It may enable a better study match with programmatic learning needs and available budget without compromising robustness.

Sample size calculation for estimating program impact requires an estimation of the treatment effect size. The treatment effect size can be understood as the difference the intervention will generate when compared to a hypothetical counterfactual such as no intervention. A small decrease in the expected treatment effect size usually means a huge increase in the sample size needed [32], and that may have important operational implications for the study operationalization and its overall cost.

The global coverage of DTP1 is estimated to be around 89% [24]. This means that in many countries, which may legitimately be concerned with ZD children in some key communities, a high national coverage level for DTP1 is also expected. Depending on the ZD distribution in the national territory, some countries may find that their key communities with higher numbers of ZD children, may already have reasonably high DTP1 coverage.

Estimating impact of programmatic approaches when the baseline is already high may be statistically challenging, as the estimated treatment effect size will also be low. For example, it may require a significantly lower sample size to statistically estimate impact when we expect an increase in coverage from 20% to 80% as compared to an increase from 70% to 80%. By including the first-year age cohort in a survey and on the ZD operational definition, we would have an increased expected treatment effect size, because the baseline would definitely be much lower. Unfortunately, untimely vaccination is much more common than missing immunization at 12 months in most settings. This may have important feasibility implications for program impact studies using targeted surveys at local levels, especially in communities with a relatively high baseline DTP1 value.

In addition to a reduced sample size need, incorporating a first-year age cohort in the survey will also have important implications related to the time it takes to demonstrate the impact of an intervention. That is because when a new immunization-related intervention is introduced, researchers may need to wait a significant amount of time to ensure the intervention will be operationally stable and the targeted population will be fully covered. It will also take some time for the results of the new program to start to appear and be adequately measured. Often, there is also uncertainty in terms of precisely when the intervention will be operationalized which may add additional delays in the study workplan. In practice, in a traditional 12–23-month cohort survey, that would mean a 3-year waiting period for measuring impact with at least 2 stable years of implementation to properly enable its documentation. That time may fall outside the evaluation funding window or the programmatic evidence need.

The addition of the first-year age cohort in these cases will enable a reduction in the time needed to follow up programmatic impact by at least one year. The detection of early effects on DTP1 coverage will be already meaningful and it will very likely translate into an effective ZD reduction later on. Although the correlation between improvements in DTP1 coverage in the first year of life and improvements in ZD results is not yet established across different contexts, it makes theoretical sense, and it is likely that DTP1 coverage in the first year of life may serve as a proxy for broader ZD impact. If the first 2 years of life are included and multiple rounds of surveys are performed, the study could also demonstrate how well reaching ZD children in a timely manner in targeted missed communities will finally translate into ZD programmatic results as defined by the global community, contributing to strengthening the evidence on measuring DTP1 immunization timeliness as a proxy indicator for ZD children. This may be a clear priority for ZD research in the coming years.

### 2.4. The Most Critical Consideration Is an Improved Utility of Evidence for Decision-Making in Rapid Learning Cycles

Finally, the overall objective of implementation research is to generate useful evidence and support decision-making of key stakeholders to improve program implementation and impact in a timely way. The inclusion of different age cohorts in the same survey may enable strengthening of this critical use case.

It will enable researchers to generate timely and meaningful insights on ZD determinants and on different issues such as timeliness of vaccination, COVID-19 impact, and dynamic shifting contexts. In particular, when the first year of life is included, it may significantly reduce the time needed from program operationalization to evidence on its impact. Through rapid learning cycles, it may equip local and national policy-makers and practitioners on current determinants communities may be facing to make timely and adequate decisions. Although less useful for international comparisons, it still could provide insights on ZD children following international standards if data can be disaggregated by different age cohorts.

Understanding coverage and determinants in the first year of life will certainly not be as useful to establish systematically missed communities or to compare with other surveys or local administrative data, but covering this age range may be critical to generate insights on the population currently being targeted by the program and how different they may be from previous cohorts. This information will better link to programmatic decision-making and enable the program to perform fine adjustments as activities are being rolled out and that could not be accomplished with a more traditional age cohort selection.

## 3. Key Decisions on Targeted Local Surveys from the ZDLH Initiative

In the case of the ZDLH initiative, the decision was made to include the age cohort from 18 weeks to 23 months, effectively including a large part of the first year of life and with a recommendation to enable disaggregation of the data across cohort years in the analysis plan. Eighteen weeks was selected as a starting date because it enables the detection of clear delays on DTP1 vaccines—for at least 12 weeks—and enables the detection of early—2 weeks—delays in DTP3. The hope is that this first-year cohort may generate early insights on ZD children and enable initial calculations of DTP drop-out rates. Twenty-three months was selected as the end date to enable comparisons with international surveys. 

Among the key reasons for the ZDLH initiative to include the first year of life included the ability to generate insights on timeliness of immunization, the ability to better identify systematically missed communities, the ability to understand programmatic performance and shifts in ZD determinants across years, and the ability to estimate program impact at an earlier stage, so that all those pieces of information can be linked to program adaptation in a timely way. Other practical reasons considered were a reduction in the sample size and time needed to estimate impact and operational simplification to answer some key questions and the associated costs for countries with higher DTP1 coverage in those communities.

Considerations was also given to including later age cohort years, especially in the context of “The Big Catch-Up initiative”, but the ZDLH group decided not to, mainly because key local questions were generally not related to this initiative, but also because some insights on COVID-19 recovery on those communities could already be generated by analyzing data from the 12–23-month cohort. Including later age cohorts was thought to significantly increase the sample size needed and the project budget without a clear use case.

## 4. Conclusions

Even though the global operational and strategic definition of ZD children for surveys is the lack of DPT1 among children aged 12–23 months, there are many reasons why different age cohorts should be included in targeted local surveys. The inclusion of the first year of life cohort may be relevant to generate useful insights on immunization timeliness, minimize recall bias, and may potentially enable the reduction in sample size and time needed to detect impact, when this is a critical research question. It may significantly improve the utility of evidence for decision-making, as insights will be generated for the population being currently targeted by the program. The inclusion of older age cohorts in the survey may also be relevant to generate insights and inform catch-up activities for older groups, but may increase recall bias. The inclusion of multiple age cohorts in the same survey may enable comparison of results across different age cohort years and support a better identification of systematically missed communities, supporting the validation of set programmatic priorities. It may also generate insights on changes in enablers and barriers to immunization under dynamic contexts such as the introduction of a new ZD intervention or when recovering from the impact of health system shocks. Including multiple age cohorts may require larger sample sizes if results need to be disaggregated by cohort years, but may enable a potential reduction in the need for household visits to find eligible children.

We believe that the approaches laid out in this article may enable better evidence and greatly contribute to improve inequalities in immunization. We think that flexibilities on the age cohort of inclusion in targeted surveys at the local level is an important principle to be considered to improve monitoring of inequalities and to respond to local ZD learning agendas needs. Rather than generating misalignments with the international definition, we think this approach may enable better, more timely and complementary data for ZD learning agendas and critically, it may position implementation research to enhance monitoring and answer learning needs in rapid learning cycles. In this sense, aligning the survey age cohort with international definitions may not be feasible or desirable. Researchers and program managers may need to consider those aspects in their decision-making when surveys are planned.

More research is needed to better understand the specific contexts where improvements in timeliness of DTP1 immunization in the first year of life will translate to improvements in DTP1 coverage in the cohort of 12–23 months as defined by the global ZD indicator.

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
