# Peer review of "Measuring Zero-Dose Children: Reflections on Age Cohort Flexibilities for Targeted Immunization Surveys at the Local Level"

_vaccines, 2024, doi:10.3390/vaccines12020195_

Round 1
Reviewer 1 Report
Comments and Suggestions for Authors
Thank you for asking me to review this article.
The theme chosen by the authors is interesting, especially in the post-pandemic period, given that in 2021 alone there were more than 14 million "zero dose" children, i.e. those who did not receive even one dose of pediatric vaccination (considering the vaccine against diphtheria, tetanus and pertussis – DTP the global indicator of early childhood vaccination coverage). This data appears even more worrying if we consider that many of these data refer to developing countries where factors such as socioeconomic deprivation and determinants such as childhood malnutrition and the weakness of the immune system expose the most fragile to the risk of infectious diseases, many of which could be easily preventable through vaccination.
However, despite the relevance of the topic, I believe that some issues need to be resolved:
The article presented by the authors is a Perspective and intends to summarize the contents of a workshop whose theme was focused on zero doses and related measurements; however the presentation of the contents, in my opinion, seems more suitable for a chapter of a book.
What is the research question? It is not evident in the abstract or even in the introduction.
Although the context is also described with specific bibliographical references, it is not clear what the purpose of the study is and what contribution the research conducted by the authors or the considerations presented can make to the literature.
The paper, moreover, is proposed under the special issue "Inequality in Immunization 2024", therefore perhaps the authors could consider delving into the topic of deprivation by dedicating a paragraph to this very current topic in relation to important health outcomes such as pediatric vaccinations.
The contents are presented too conversationally, which distracts the reader. Perhaps the authors could consider discussing the contents starting from the description of the context and then commenting on any strengths and/or weaknesses of the measurements currently under consideration, highlighting any critical issues and discussing useful proposals in view of future perspectives.
Author Response
Thank you very much for your timely and relevant review. This is much appreciated. We agree with the importance of the theme and got positive feedback for the draft of this article from multiple reviewers across different Alliance partner organizations before we submitted it to Vaccines.
We agree on the importance of feeding this information in more formal guidelines and we are pushing for its contents to feed into current WHO guidance discussions, but we respectfully disagree that the contents would be a better fit for a book chapter. The IA2030 strategy is getting closer to its mid-period in an exceptional post-pandemic setting which already imposed delays in its operationalization. Multiple partners agree that there is some urgency on reviewing global guidance to make targeted local surveys / assessments better fit for purpose for the ZD agenda. WHO is working already on guidance for ZD assessments and has appreciative our earlier drafts. We hope this perspective will contribute to this global debate in a timely and relevant way and we believe this message needs to be available for a broader audience working with ZD children and implementation research in immunisation as soon as possible.
Because this is a perspective article and not an original research article, we don`t have a research question. However, we do have a strong rationale for suggesting this perspective article which was laid out in the introduction. We tried to rewrite the objective paragraph to make it even clearer.
In sum, it suggests methodological perspectives for targeted surveys at local level to make them better fit to answer equity related learning questions and feed into decision making in rapid cycles. Current global survey guidance focuses mainly on global partner needs without a deeper consideration on the diverse needs of local learning agendas, the debate is particularly poor when considering different age cohorts of inclusion. We believe this rationale was now made sufficiently clear in the article.
Although multiple deprivations were not the key focus of the workshop discussion, it is implicit that local surveys can collect relevant data from specific communities related to this relevant theme, but we agree we could make a better job in highlighting this relevant area. We added some language related to that across the document and reworded some specific paragraphs in the introduction. Thanks for this suggestion. We added some wording and references to multiple deprivations and equity relevance in the text from lines 101 to 129.
We did write with simple language and provided multiple examples to make a difficult methodological discussion more palatable for a wider audience, but we agree we could make key messages clearer. We rewrote the conclusions to better summarise what we discussed with strengths and weaknesses of each alternative approach clearly highlighted.
Reviewer 2 Report
Comments and Suggestions for Authors
Comments for authors
- Lines 40-42: Revise for clarity.
- Abstract: The authors emphasised the need for flexibility but did not present their perspective/suggestion on the age cohort for measuring Zero-dose children
- Introduction: The authors need to start the introduction with a clear definition or description of Zero-dose children.
- Lines 70-72: Could you provide a citation to back up these claims?
- Line 157 and above: Please number the sections appropriately
- Reference: Format the references as per the journal style
- Considering the affiliations of the authors, they should declare any potential conflict of interest in the manuscript.
Author Response
Thank you very much for your very useful and clear review. We appreciate your time and effort to improve our article. See responses to your points below.
- Line 40-42 – we removed this sentence for better clarity.
- The main point of the article is that flexibilities are needed to adapt to local learning needs. In this sense it is hard to present a suggestion that may fit in different needs and use cases. Having said that we did present the age cohort selected by the ZDLH group in the abstract and we are further emphasising the critical advantages of including the first year of life. In addition, we rewrote the conclusion to better highlight advantages and disadvantages of each approach.
- The Zero Dose definition has been added upfront.
- The citation is number 7 Scobie, H.M.; Edelstein, M.; Nicol, E.; Morice, A.; Rahimi, N.; MacDonald, N.E.; Carolina Danovaro-Holliday, M.; Jawad, J. Improving the Quality and Use of Immunization and Surveillance Data: Summary Report of the Working Group of the Strategic Advisory Group of Experts on Immunization. Vaccine 2020, 38, 7183–7197, doi:10.1016/J.VACCINE.2020.09.017. , which was closer to the denominator problem sentence, but referred to both numerators and denominators problems. We added the two sentences with an “and” to avoid confusion.
- The original submitted version in docx had a correct numbering of section. There might have been issues during the formatting from the journal side. We corrected the numbering in this last version according to the original submission.
- This has been done.
- A “conflicts of interest” section has been added.
Reviewer 3 Report
Comments and Suggestions for Authors
The article offers a comprehensive overview of considerations for employing diverse age cohorts in localized immunization surveys; however, there is a need to enhance the introduction by clearly articulating the article's primary focus. Specify whether it primarily serves as a methodology guide, explores learning needs, or examines the ZDLH initiative's decisions. This clarification will set more explicit expectations for readers.
Authors are suggested to provide a concise summary of key details related to the ZDLH initiative in the introduction or methods section to establish context. Include information on the countries involved, objectives, and types of targeted surveys employed.
While the conclusions effectively summarize the ZDLH decision, consider expanding to underscore broader principles and considerations for researchers or programs contemplating age cohort flexibility in targeted surveys. Authors can also offer guidance on when and how to implement such flexibility.
Comments on the Quality of English LanguageThe article is generally clearly written but has some minor grammatical errors that should be addressed.
Author Response
Thank you very much for your useful and relevant review. We addressed your points in the text and provide some answers to your queries below.
The primary focus of the article is to generate insights to improving methodological approaches for targeted surveys at local level to make them more equity relevant and better fit for informing learning agendas. We reviewed the introduction to make it clearer.
We believe we provided enough details of the ZDLH initiative in the article, and we would prefer not to expand too much to not lose focus on the key issue being discussed. Countries involved are already stated in the text and the key objectives of targeted local surveys as well. But we added a reference to the ZDLH web platform in case readers would like to get more details on the initiative and the survey approaches from different countries.
We expanded the conclusion to underscore broader principles and considerations for researchers or programs contemplating age cohort flexibility.
We also revised the article for grammatical errors and hope they have all been addressed.
Round 2
Reviewer 1 Report
Comments and Suggestions for Authors
The authors responded thoroughly to my comments. They have expanded on some aspects of the introductory section and described the conclusions more clearly.
I believe the work can be published in its current form.
Author Response
Thank you very much for your review! Your contribution to our work is deeply appreciated!